# In-context Curriculum for Mathematical Reasoning in Small Language Models

## Abstract

Specializing Small Language Models (SLMs) in mathematical reasoning improves the scaling of model performance and reduces the cost of inference. Leveraging the model's context is key for specialization and parameter-free adaptation in the In-context Learning (ICL) paradigm. In the case of Large Language Models (LLMs), more reasoning steps in the chain-of-thought (COT) based demonstrations within an ICL prompt are known to result in higher accuracy during testing on mathematical reasoning datasets such as GSM8K. Although SLMs have limited capability for multi-step COT reasoning, prior works in specializing SLMs use multi-step COT-based demonstrations to encapsulate model context. We propose an alternative termed the In-context Curriculum Random (ICCR) prompt which varies the complexity of demonstrations, ranging from a simple single COT-based reasoning step to more complex multi-step COT-based demonstrations. ICCR achieves a 16.15% inference accuracy on the GSM8K dataset, surpassing the 14.33% accuracy displayed by the GPT 3.5-distilled COT baseline for SLM specialization. Unlike the aforementioned baseline, ICCR uses out-of-distribution datasets, i.e., ASDiv, SVAMP, and MathQA, which serve to emphasize simpler COT-based reasoning prompts. In the context of ICL, basic arithmetic calculation-based demonstrations in a natural language format are shown to outperform both the baseline and ICCR prompts on the Google FLAN-T5 XL and XXL models. We conclude that at model scales from 250M to 11B parameters, simpler COT-based reasoning prompts result in higher performance.

## 1 Introduction

The context in which a language model operates is crucial for its ability to generalize across tasks and to adapt to new information over time, as highlighted by (Dong et al. (2022)). Interestingly, this context can be leveraged to train language models to perform new tasks without even modifying their underlying parameters, a process known as In-context learning. In-context learning (ICL) takes demonstrations and optional instructions as input in order to elicit desired performance on an unseen task without changing the weights of a language model. It does so by adding these demonstrations as the context, also known as "prompt" in a language model. ICL and math reasoning are considered emergent abilities that only language models at a certain scale exhibit. However, there is a study that emphasizes emergent abilities as a result of continuous improvement, only visible at a certain scale due to discontinuous metrics and tasks (Schaeffer et al. (2023)). Perhaps, it's possible to learn emergent abilities such as mathematical reasoning at a smaller scale. Specializing small language models on mathematical reasoning is an important problem that can reduce inference time costs of language models and provide efficient scaling of performance in a resource-constrained setting (Fu et al. (2023); Magister et al. (2022a)). A foundational study we reference as our baseline employs the Chain-of-Thought reasoning prompts, designed in the In-context learning format, to specialize small language models (Fu et al. (2023)). So, what does "specialized" mean in this context? Specialization is a process that merges traditional deep learning training techniques, where a model is honed to its minimal validation loss, with In-context Learning (ICL) prompts. This synergy enables the model to redirect its general language modeling prowess to focus on and excel in the specific task it's being trained for.

Chain-of-thought (COT) reasoning prompting (Wei et al. (2022b)) has been a highly effective method in eliciting reasoning in large language models. Simply asking a language model to think

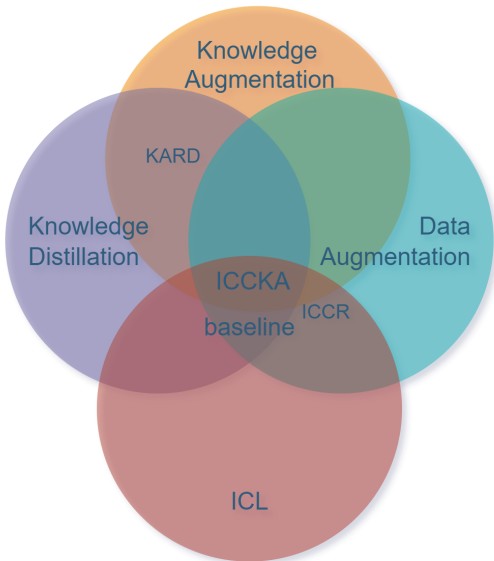

Figure 1: This figure shows how baseline (Fu et al. (2023)), knowledge augmentation method called KARD (Kang et al. (2023)), and our work of ICCR and ICCKA on specialization fit into four key fields of Knowledge distillation, knowledge augmentation, ICL, and data augmentation

step by step elicits COT in answers. Combining COT with ICL has become a standard practice in reasoning tasks where demonstrations showing their reasoning steps enable better adaptation of language models to reasoning tasks. A key finding in ICL prompts with chain-of-thought (COT)-based reasoning shows that greater prompt complexity, i.e., more reasoning steps in a COT-based demonstration prompt improves the model performance (Fu et al. (2022)) on difficult reasoning tasks such as GSM8K (Cobbe et al. (2021)). Prior works that specialize small language models (Fu et al. (2023); Magister et al. (2022b); Kang et al. (2023)), use COTs to encapsulate context in small language models. Given the limited COT and in-context learning ability in small language models, we question whether the premise of higher-complexity ICL prompts resulting in higher performance applies to models at smaller scales. Therefore, to improve the accuracy of specialized small language models, we propose an intuitive strategy that organizes prompts in the form of a curriculum.

We introduce the In-context Curriculum (ICC) prompt, designed to mirror the structure of a curriculum. It begins with simple demonstrations, primarily characterized by one-step Chain of Thought (COT) solutions, emphasizing straightforward reasoning. As the prompt progresses, the complexity increases, culminating in a minority—less than 25%—of demonstrations that feature multi-step COTs. These multi-step solutions represent more intricate reasoning processes, highlighting the gradual escalation from simplicity to complexity within the ICC as shown in Figure 2. Our method bears resemblance to the least to most prompting method that creates multiple questions from each step of multi-step COT and proves highly effective on math reasoning tasks (Zhou et al. (2022)). We conduct our experiments on the GSM8K dataset (Cobbe et al. (2021)), which contains around 7290 multi-step math word problems for training. We adopt the methodology outlined in Fu et al. (2023), which specializes in small language models to tackle math word problems through knowledge distillation and In-context Learning (ICL). Knowledge distillation involves a larger, more knowledgeable "teacher" model imparting its insights to a smaller "student" model. Specifically, the GPT 3.5 model by OpenAI (Brown et al. (2020)) serves as the teacher, generating Chain of Thought prompts for the GSM8K training set. These COT prompts are then employed as ICL prompts during training. This approach forms our baseline, against which we evaluate the effectiveness of our ICL prompt called In-context Curriculum (ICC).

We leverage knowledge augmentation and data augmentation within our work. Both augmentations are similar in their use of external data, however, differ mainly in retrieval and position of usage. Knowledge augmentation retrieves knowledge that is relevant to the query at training and inference.

The retrieved information might be given to hidden states of the model or their input or output labels. In our case, we use retrieved knowledge to design prompts for our In-context Curriculum Knowledge augmentation (ICCKA) method. Data augmentation provides external data as pairs of input or output for training, rather than on hidden states. In our work, we use data augmentation to simply construct our ICL prompts for the In-context Curriculum Random (ICCR) method. We show how our work of ICCKA and ICCR fit into discussed fields in Figure 1. Knowledge distillation uses external data and can be considered as a form of knowledge augmentation, but from an external model rather than a source of data. Both ICCKA and ICCR follow the same structure as outlined in figure 2.

We undertake two primary experiments on the GSM8K dataset using our ICL-style prompts 1) Specializing the Google Flan-T5 base. 2) Implementing In-context Learning (ICL) on the Flan T5 family (Chung et al. (2022)), which involves parameter-free adaptation. Building on these experiments, our primary contributions are as follows:

- **Curriculum style prompt outperforms Chain-of-thought** We introduce the In-context Curriculum Random (ICCR) prompt, which outperforms traditional Chain-of-Thought prompts for the specialization of small language models

- **Efficacy of Basic Arithmetic Demonstrations in ICL:** In the ICL domain, our "basic arithmetic Q/A" demonstrations, based on straightforward arithmetic calculations, consistently excel over both ICL Chain-of-Thought and ICCR prompts for the Flan-T5 model range.

- **Knowledge Augmentation Evaluation:** By comparing ICCKA with ICCR prompts on the GSM8K dataset, we highlight the limited effectiveness of Knowledge Augmentation across specialization and ICL tasks for small language models.

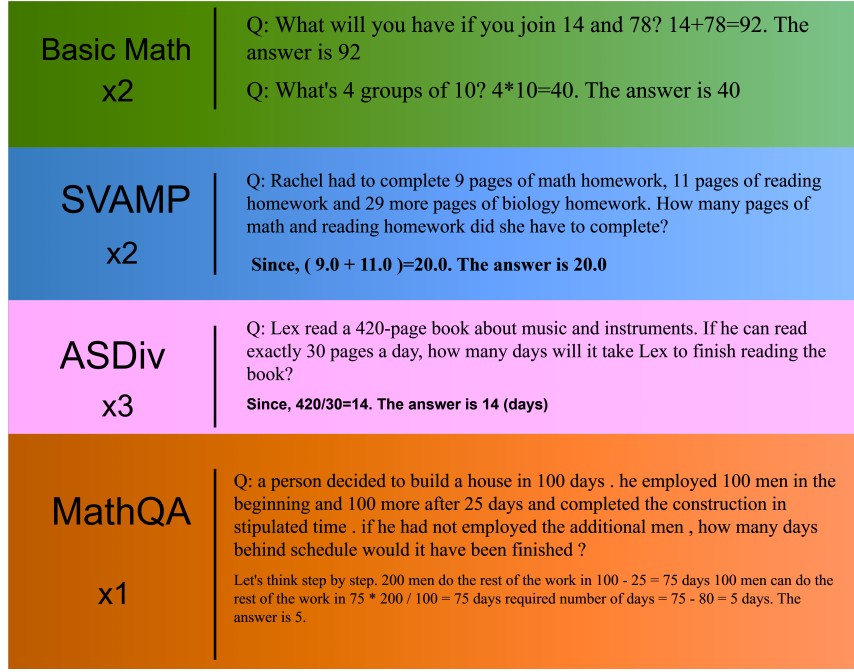

Figure 2: This figure illustrates the composition of our ICC prompts: it begins with 4-5 basic arithmetic Q/A demonstrations, followed by 2 SVAMP, 2 ASDiv, and concludes with 1 MathQA demonstration.

## 2 BACKGROUND

Specializing small language models is a resource-constrained way of finding better ways to scale the performance of LLMs in one domain. One common way (Magister et al. (2022b)) is to teach a small language model using knowledge distillation from a large language model that augments data by providing more detailed ground labels in the form of Chain-of-thought (COT) reasoning steps. Knowledge distillation from 540B PaLM model has been proven to teach a small language model with 770M parameters to achieve similar performance across 4 NLP benchmarks (Hsieh et al. (2023)). Specialized Small language models for math reasoning (Fu et al. (2023) ), that we use as a baseline, implement knowledge distillation by distilling knowledge from GPT 3.5(Brown et al. (2020)) using COT ground truths. The generated COT ground truths are also used in prompts as ICL demonstrations to better guide the specialization process. They show how the specialization process is achieved by trading generic language modeling abilities for specific abilities like math reasoning.

Combining external knowledge with language models can be done using prompts as is standard practice in knowledge augmentation (Kang et al. (2023)). Giving useful external knowledge within prompts is a safe strategy across a wide set of tasks. As discussed in Introduction 1, knowledge augmentation through prompts can be done by retrieving relevant information. Since math word problems are distinct reasoning challenges that don't rely on domain-specific facts for correct answers, we utilize demonstrations. These are presented as ICL examples to augment knowledge. The field that concerns itself with finding the best selection of prompt examples in solving a task is In-context Learning (ICL) (Dong et al. (2022)). In recent years, the study of In-Context Learning (ICL) methods on large language models has garnered significant attention. There's a general consensus in ICL that examples of similar or greater complexity compared to target problems yield the best results for large language models, especially when related to their COT demonstrations(Fu et al. (2022)). There is limited study of ICL on small language models due to their low performance (Wei et al. (2022a)). However, there is reason to believe that small language models's ICL ability, albeit in a low-performance zone presents behavior similar to larger scales. This perspective suggests that emergent abilities are present even at lower scales, but they may appear absent due to inconsistencies in measurement metrics (Schaeffer et al. (2023)). One example of ICL's influence in smaller scales is the observation that in-context data helps generalize small language models across zero-shot tasks and improves performance during specialization (Fu et al. (2023)). Our work corroborates that context from demonstrations affects the specialization process, as well as the ICL performance in small language models.

In order to find the best ICL prompts for the specialization of small language models, we investigate the ICL literature that relates to the complexity of Chain-of-thought (COT) prompts. Some works indicate the promise of using problems of differing complexity within the context (Zhou et al. (2022); Ye et al. (2023)). Complexity relates to the hardness of the problem and the number of steps it requires to solve it. For Least-to-Most prompting (Zhou et al. (2022)), authors show that COTs can be broken down into sub-problems that are related to the sub-step. Another similar work poses ICL as a subset selection problem and shows a compositional example building that shows promise for a curriculum-like structure (Ye et al. (2023)). The underlying mechanisms of ICL remain an enigma, as highlighted in (Dong et al. (2022)). Interestingly, research indicates that the traditional ground truth labels may not always enhance ICL performance. For instance, substituting standard labels in sentiment analysis with arbitrary labels like "foo" and "bar" yielded superior outcomes, suggesting a shift towards problem generalization rather than pattern memorization (Wei et al. (2023)). Another study echoed this sentiment, revealing that replacing ground truth labels in reasoning tasks had a negligible impact on performance (Min et al. (2022)). In the realm of CoT-based examples, labels are not entirely counterproductive; their influence, however, is minimal, with the structure and relevance of CoT emerging as more pivotal factors (Wang et al. (2022)). The pivotal aspects of CoT demonstrations were found to be the structure of CoT and its relevance. Collectively, these findings challenge the conventional understanding of ICL. Rather than instructing language models through example demonstrations, ICL appears to facilitate implicit Bayesian inference, where models discern a latent concept shared among examples in a prompt (Xie et al. (2021)).

## 3 METHOD

Our main contribution is designing an In-context Learning (ICL) prompt,

$$C = \{I, s(x_1, y_1), \ldots, s(x_k, y_k)\} \tag{1}$$

that follows a curriculum from out-of-distribution datasets as shown in Figure 2 where $s(x_k, y_k)$ is one example. We introduce three new ICL prompting methods named In-context Curriculum Random (ICCR), In-context Curriculum Knowledge Augmentation (ICCKA), and Basic arithmetic Q/A. Both In-context curriculum methods add external data containing math word problems as prompts by replacing ICL prompts formed with GSM8k COT demonstrations. Our In-Context Curriculum Random (ICCR) prompts can be considered data augmentation which changes the ICL demonstrations that our models are trained and tested on. Three external math word problem datasets used are SVAMP, ASDiv, and Math-QA (Patel et al. (2021); Miao et al. (2020); Amini et al. (2019)). Unlike the baseline we enhance, all our prompts are sourced from Out-of-distribution (O.O.D) datasets. IC-CKA contains examples most semantically similar to our target query. We show how our work of IC-CKA and ICCR fit into discussed fields in Figure 1. Our In-context Curriculum (ICC) prompt is designed to mirror the structure of a curriculum. It begins with simple demonstrations, primarily characterized by one-step Chain of Thought (COT) solutions, emphasizing straightforward reasoning. As the prompt progresses, the complexity increases, culminating in a minority—less than 25%—of demonstrations that feature multi-step COTs. These multi-step solutions represent more intricate reasoning processes, highlighting the gradual escalation from simplicity to complexity within the ICC as shown in Figure 2. Our method bears resemblance to the least to most prompting method that creates multiple questions from each step of multi-step COT and proves highly effective on math reasoning tasks (Zhou et al. (2022)).

### 3.1 BASIC ARITHMETIC Q/A

At the beginning of the figure 2, you can see an example of basic arithmetic Q/A. This is the data that is generated by GPT-4OpenAI (2023). We prompted GPT4 through ChatGPT to give 10 examples for each operator: addition, subtraction, multiplication, division, and percentage. We asked it to generate "unique templates involving natural language for each operator". We use a random selection of these for both ICCR and ICCKA prompts.

### 3.2 ICCR: IN-CONTEXT CURRICULUM RANDOM

In ICCR prompts, we do a random selection from the list of datasets. We don't use information relating to the problem and its answer. The in-context prompt is populated with demonstrations up to a 2500-character limit, but not exceeding 12 demonstrations. This leads to a random selection from each dataset, with ASDiv getting the highest priority and then SVAMP. We give the least importance to MathQA, which contains Chain-of-thought explanations and matches our target distribution. We chose this approach, believing that simpler examples would be the most effective curriculum for the small language model, especially considering its limitations in processing Chain-of-thought compared to larger models, as indicated by the (Dong et al. (2022)). Recognizing the value of complexity, we ensured the inclusion of at least one example from the MathQA dataset, which features CoT with multi-step solutions, to help the model gradually reach that level of reasoning.

### 3.3 ICCKA: IN-CONTEXT CURRICULUM KNOWLEDGE AUGMENTATION

In this format, the prompt is constructed based on the query and answer keys. The predominant operator identified from the answer key dictates the type of in-context demonstrations, favoring examples that utilize the same operator. We employ a sentence transformer to generate embeddings for queries from GSM8k and all out-of-distribution (OOD) datasets. Depending on the query, we retrieve the most pertinent problems associated with a specific operator. Both ASDiv and SVAMP datasets offer operator-based tags for this purpose. However, for MathQA, we simply query from the whole dataset to find the most relevant questions. The outline of our process that constructs an ICCKA prompt is shown in Figure 3.

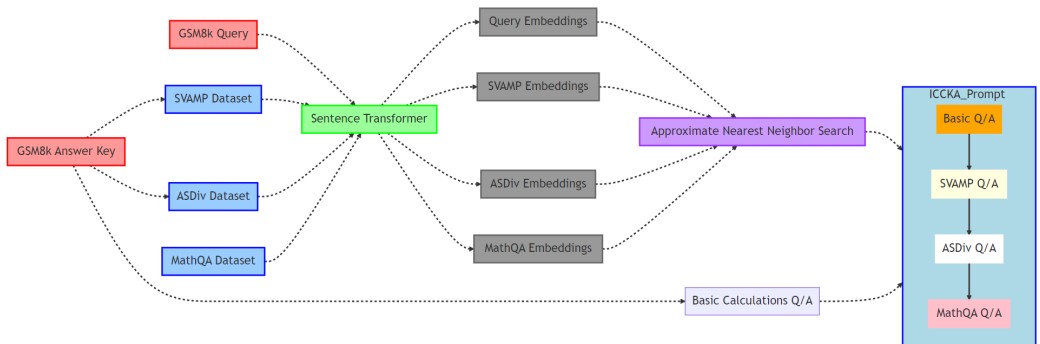

Figure 3: This figure shows the pipeline of an ICCKA prompt constructed for a query and answer from the GSM8K training set. We use knowledge augmentation by first generating embeddings with sentence transformer (Reimers & Gurevych (2019)) and then we use approximate nearest neighbor search to find the most semantically similar questions to the given query (Johnson et al. (2019)). The answer key is used to make an operator-based filtered search.

## 4 EXPERIMENTAL SETUP

We conduct our experiments on the GSM8K dataset (Cobbe et al. (2021)), which contains around 7290 multi-step math word problems for training. We adopt the methodology outlined in (Fu et al. (2023)), which specializes in small language models to tackle math word problems through knowledge distillation and In-context Learning (ICL). Knowledge distillation involves a larger, more knowledgeable "teacher" model imparting its insights to a smaller "student" model. Specifically, the GPT 3.5 model by OpenAI (Brown et al. (2020)) serves as the teacher, generating Chain of Thought (COT) prompts for the GSM8K training set. These COT prompts are then employed as ICL prompts during training. This approach forms our baseline, against which we evaluate the effectiveness of our ICL prompt method called In-context Curriculum (ICC).

### 4.1 SPECIALIZATION

We have enhanced the specialization process of the Google Flan-T5 base model using our In-context Curriculum prompts. We substitute the in-context chain-of-thought training data, which comprises approximately 200,000 entries. Given that the training data encompasses around 7,290 problems, we construct 200,000 problems by repeating them, ensuring each repetition has a unique in-context prompt. Our prompts modify the in-context data's nature, maintaining the knowledge distillation component. We train the model using the same labels as the baseline, ensuring that each repeated problem has a distinct prompt. During inference for ICCR, we select the best out of three prompts to generate test results. In contrast, for ICCKA, we retrieve the most semantically similar prompts based on the query from our dataset collection.

To produce 200,000 distinct entries for ICCKA, we sampled up to 50 repetitions of the same problem, combined with the nearest neighbors of the query. While these combinations were unique, they did include repetitions. This approach contrasts with ICCR, where repetitions are entirely random, rather than being semantically related.

We trained the Google FLAN-T5 base model for 10 epochs using the prompts crafted for both ICCKA and ICCR. Post-training, we performed inference on the trained model using our in-context prompts.

### 4.2 IN-CONTEXT LEARNING

In the realm of in-context learning, we evaluated the efficacy of our in-context prompts across the Google FLAN model spectrum, from the large to the XXL model. We employed the best of three prompts for each category, with the exception of ICCKA. For ICCKA, we retrieve the most seman-

Table 1: This table compares ICCR, ICCKA, and baseline ICL prompt on the specialization of Google Flan T5 Base. Column names indicate the type of prompt used to train the SLM. And rows indicate the ICL prompts used during inference.

| Inference Type | Baseline | ICCR | ICCKA |
|---|---|---|---|
| baseline | 0.1433 | 0.1562 | 0.1440 |
| ICCR | 0.1569 | **0.1615** | 0.1387 |
| ICCKA | 0.1554 | 0.1516 | 0.1478 |
| no prompt | 0.1440 | 0.1516 | 0.1448 |

Table 2: This table breaks down different components of the ICCR prompt from Table 1 and measures their effect on final results.

| | basic Q/A | Svamp IC | AsDiv IC | MathQA IC | ICCR-Random order | ICCR-ordered |
|---|---|---|---|---|---|---|
| **Baseline** | 0.1478 | 0.1516 | **0.1660** | 0.1403 | 0.1547 | 0.1585 |
| **ICCKA** | 0.1471 | 0.1433 | 0.1418 | **0.1516** | 0.1433 | 0.1387 |
| **ICCR** | 0.1516 | 0.1577 | 0.1486 | 0.1531 | 0.1478 | **0.1615** |

tically similar prompts based on the query from our dataset collection. This is exactly the same approach we took during inference on Specialized models.

## 5 RESULTS

We use a straightforward accuracy metric, that measures the ratio of correct to incorrect problems. For example, an accuracy ratio of 0.1577 means that the model correctly predicted 208 problems out of 1319 test problems. We simply refer to it as the accuracy.

### 5.1 SPECIALIZATION

In table 1, each column represents the training method and each row represents the inference method. We get the best results with a model trained with ICCR data and tested on ICCR prompts. Moreover, ICCR prompts are able to reproduce most of their performance without training on ICCR prompts for the baseline model. We believe that this shows the efficacy of our prompt. These results also indicate how the multi-step COT in context can be replaced to give better results with simpler questions. ICCKA performs the worst and we believe it's because of an effect noticed in Wei et al. (2023) where more semantically relevant retrievals come in the way of generalization. Another reason could be the lack of symmetry in inference and training during ICCKA. We did not use answers to guide prompts in testing data. Many works such as Kang et al. (2023) created learned rankers for inference, but we refrained from doing that step as we wanted to focus on understanding ICL better with our experiments. We use Basic arithmetic demonstrations at the beginning of both ICCR and ICCKA prompts. We find that its usage marginally improves the adaptation of specialized Language models. For instance, our ICCR prompts without basic Q/A averaged slightly below 0.16 accuracy.

### 5.1.1 ABLATION STUDIES

We analyze the effect of each component of the ICCR prompt and its order in Table 2. Our findings suggest that the order of ICCR components plays a pivotal role, albeit to a certain degree. Interestingly, while individual prompts don't necessarily boost performance on their own—with the exception of ASDiv—their synergistic combination when sequenced correctly, yields superior results.

Table 3: In-context Learning on all our designed Prompts

| ICL Prompt | google-flan large | google-flan xl | google-flan xxl |
|---|---|---|---|
| **Baseline** | 0.0675 | 0.1099 | 0.1577 |
| **ICCR** | 0.0690 | 0.1077 | 0.1592 |
| **ICCR-random order** | 0.0720 | 0.1039 | 0.1676 |
| **ASDiv** | 0.0728 | 0.1092 | 0.1744 |
| **basic Q/A** | 0.0713 | **0.1138** | **0.1759** |
| **SVAMP** | 0.0652 | 0.1054 | 0.1721 |
| **Math QA** | **0.0781** | 0.1039 | 0.1749 |
| **ICCKA** | 0.0720 | 0.1046 | 0.1713 |

This underscores the importance of a well-structured curriculum design. The standout performance of ASDiv can potentially be attributed to its instrumental role in the formulation of GSM8K Cobbe et al. (2021)

### 5.1.2 IN-CONTEXT LEARNING ON GOOGLE FLAN FAMILY

We evaluate the ICL ability of all our prompts, including basic arithmetic Q/A in Table 3. Interestingly, baseline and ICCR performance in this domain are almost equivalent. We don't see any benefit of curriculum design. In fact, sometimes we witness the detrimental effect of the curriculum order on performance. Most of our designed prompts feature individual data sources. We find an overarching pattern that the simple question prompts like ASDiv and basic arithmetic Q/A outperform baseline, ICCR, and ICCKA. We see that basic arithmetic Q/A performs the best on Google Flan XXl 11b and Flan Xl 3B. Whereas, results are much more random for FLAN Large 780M. We fail to see any pattern in the Flan Large model. It might be because of the model's extremely low performance on this dataset.

## 5.2 ANALYSIS OF RESULTS

### 5.2.1 SPECIALIZATION ON GOOGLE FLAN-T5 BASE

Our results demonstrate the efficacy of ICCR prompts on the specialization of SLMs and demonstrate how ICL prompt complexity influences SLM performance. The efficacy of an intuitive strategy such as ICCR indirectly corroborates the claim that small language models do possess ICL ability, albeit one that is limited. Our work does not agree with the findings that complex ICL prompts improve the accuracy of language models on reasoning tasks (Fu et al. (2022)). We argue that the ICL prompt complexity has to be considered in conjunction with the model scale at which we are conducting experiments. We conjecture that simpler, single-step COT prompts, carry more information for language models at a smaller scale as these models make simpler mistakes. The proposed ICCKA scheme shows the ineffectiveness of knowledge augmentation on the task of math word problem specialization. We argue that it is because the ICCKA prompt causes interference with semantically similar worded problems that do not share the same structure and relevance. One work that aims to demystify ICL and COT-based prompts shows that the relevance of examples in the prompts as well as their structure is a more important predictor of their performance on reasoning tasks than their specific labels or the correctness of their labels (Wang et al. (2022)). For example, a problem involving mathematical reasoning that contains the word "birds" may have ICL prompts comprising every conceivable question about birds. This may confuse the language model into thinking that the word "birds" holds special relevance in the mathematical reasoning employed in solving the underlying problem (which it does not).

### 5.2.2 IN-CONTEXT LEARNING

We corroborate the findings in the least-to-most prompting strategies employed in (Zhou et al. (2022)) that ICL prompt complexity can be decoupled from questions posed within the prompts. We conjecture that simpler prompts are much more effective in the specialization of SLMs since they provide better feedback to the SLM's potential missteps which are simpler in nature (Uesato et al. (2022); Lightman et al. (2023)). That could also explain why LLMs are able to solve easier problems on account of the better feedback provided by more complex COT-based prompts. Our use of the word "feedback" suggests an alternative view of ICL, i.e., as a problem of providing the most appropriate feedback in light of future mistakes made by the model. Unlike previous works (Min et al. (2022)), we observe that the use of out-of-distribution (OOD) datasets in ICL provides superior performance for SLM specialization. This signifies a marked change in ICL behavior witnessed at smaller model scales versus ICL behavior observed at larger model scales. In summary, small-scale FLAN models are observed to perform better with (a) simpler COT-based prompts compared to more complex ones, and (b) prompts derived from OOD datasets than those derived from within-distribution datasets. Both of the aforementioned observations are in direct contrast to the corresponding observations made in the case of LLMs where complex COT-based prompts and prompts derived from within-distribution datasets are seen to result in improved LLM performance.

## 6 CONCLUSION

The success of strategies like ICCR and basic arithmetic demonstrations supports the idea that small language models have some ICL capability, though it's limited. In essence, small-scale FLAN models perform better with simpler COT-based prompts and those from out-of-distribution datasets. This contrasts with large language models, which benefit from complex COT prompts and in-distribution dataset prompts.

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

## A APPENDIX

You may include other additional sections here.

