# OpenReview forum: "In-context Curriculum for Mathematical Reasoning in Small Language Models"
_ICLR.cc/2024/Conference — ICLR 2024 Conference Withdrawn Submission_

### Official Review · Reviewer_LFaS · 2023-10-28

**Soundness:** 2 fair
**Presentation:** 2 fair
**Contribution:** 2 fair
**Rating:** 3
**Confidence:** 3

**Summary:**

Context is crucial for the adaptability of language models. This adaptability can be enhanced without changing model parameters through In-context Learning (ICL), which incorporates demonstrations and instructions as input. This paper aims to investigate the possibility of harnessing emergent abilities, like mathematical reasoning, in smaller models. The Chain-of-Thought (COT) reasoning is highlighted as an effective method for larger models, but its applicability to smaller models is questioned. The paper introduces the In-context Curriculum (ICC) prompt, which resembles curriculum learning. Experiments are based on the GSM8K dataset, with knowledge distillation from larger models (like GPT 3.5) forming the baseline.

**Strengths:**

The paper focuses on an intriguing and timely problem: how to harness the emergent abilities typical of Large Language Models (LLMs) in more resource-efficient Small Language Models (SLMs). If that was feasible, and if the paper had proposed an effective way to achieve that goal, it would be a significant contribution.

**Weaknesses:**

The proposed method is not at the technical level/depth that we expect from an ICLR paper.  The method simply uses some out-of-distribution datasets to strengeth the performance of in-context learning. There is nothing fundamentally new in this idea. Plus it is very unclear how these methods generalize to other problem domains.

**Questions:**

To provide a significant research contribution, the paper will need to be extended in some major ways:
- make the problem more general -- not so specific to the GSM8K dataset and the other datasets being used.
- increase the depth and novelty of the proposed method. In its current formulation, the proposed method does not seem any more sophisticated than standard prompt engineering, often described in blogs and other online resources.
- make sure that the experimental results are reproducible by others.

---

### Official Review · Reviewer_VGtM · 2023-10-29

**Soundness:** 2 fair
**Presentation:** 1 poor
**Contribution:** 1 poor
**Rating:** 3
**Confidence:** 4

**Summary:**

This paper proposes a prompting method for smaller language models for mathematical reasoning. The authors propose and compare two methods, one based on a static prompt that contains examples of increasing complexity from other datasets (ICCR), and a second one based on dynamically selected examples (ICCKA). Experiments performed on GSM8k show mixed results.

**Strengths:**

The paper tackles an interesting and current problem, of making small language models perform complex multi-step reasoning, that is well situated in the context of prior work (e.g. Tiny Stories, and Fu et al's ICML 2023 paper that is cited here).

The idea of building a "curriculum" of in-context examples might be new, as far as I'm aware. I also haven't seen exploring fine-tuning and in-context example selection together.

**Weaknesses:**

The presentation of the method is a bit vague. I had to re-read several times, and I'm not sure I understand what exactly happens in ICCR and ICCKA. For ICCR, there seems to be some dataset weighing mechanism that is not fully described. For ICCKA, the example selection method has little detail in Section 3.3.

The evaluation in the paper is extremely narrow: only T5 models (fine-tuning only T5 base, while prompting the larger T5 models), and only on GSM8k. Moreover, the effects obtained on GSM8k are also rather small (1-2 absolute percent in the best cases, with mixed results everywhere). This makes it quite hard to tell whether even these gains are in fact caused by the intuitions given in the paper (e.g., the idealized idea of a curriculum), or random low-level effects from the combination of T5, GSM8k and the prompts used.

There is very limited discussion on what qualitatively happened in the experiments. Given the small absolute results, it might indeed be hard to find examples that illustrate "why" the method might help in some cases.

Overall, I unfortunately do not know what to take away from the paper, and I'm not convinced most readers in ICLR would take away a clear message either from the ideas or from the results here. Since the results were marginal and the idea is quite specific, I'm not sure what generalizes here.

**Questions:**

- For ICCR, what exactly is the weighing of each dataset? You mention that MathQA has the least importance -- what importance is given to it, and why, since it seems like it has the most relevant CoT examples?
- For ICCKA, the paper mentions that it selects "favoring examples that utilize the same operator". Is this an explicit heuristic, or is it just an observation of what empirically tends to happen?
- What sentence embedding model was used to select examples?
- Are there any representative examples showing how the behavior of T5 changes with your prompt? These would be useful to include in the Appendix, which is currently empty.
- What are the most comparable results from Fu et al, 2023? It would help to have them side-by-side in the paper. Note that their paper used code-davinci-002 as the teacher model, which seems to not be the case here.

---

### Official Review · Reviewer_Rowt · 2023-11-01

**Soundness:** 3 good
**Presentation:** 3 good
**Contribution:** 2 fair
**Rating:** 3
**Confidence:** 4

**Summary:**

This paper proposes a new way to design in-context examples, particularly for mathematical reasoning using small language models. By varying the complexity of demonstrations (from a simple single reasoning step to more complex multi-step reasoning examples), the proposed method aims to provide a comprehensive demonstration and knowledge to the small LLMs. Empirical evaluations show that the proposed in-context examples improve the performance of small language models on mathematical reasoning tasks.

**Strengths:**

The idea is elegant and reasonable. There have been many studies on in-context example selections to improve the performance of language models. In comparison, this paper proposes a new perspective of in-context example selection: to include comprehensive different in-context examples that cover different kinds of knowledge (from simple to complex).

**Weaknesses:**

1. **Limited applicable scenarios.** It is a good idea to decompose in-context examples into different complexity levels to provide more comprehensive knowledge for small language models. However, the proposed method only covers mathematical reasoning tasks, where it is relatively easy to decompose in-context examples into multiple complexity levels. It would be important to discuss how to extend the proposed method to other tasks.
2. **Only applicable to small language models.** The proposed method aims to provide comprehensive knowledge of small language models to adapt them for mathematical reasoning tasks. However, considering the LLMs already have enough domain-specific knowledge in terms of math, the proposed method might be ineffective with LLMs. A better way to demonstrate the effectiveness of the proposed method would be testing LLMs on a specific domain where even LLMs are not good at. In that case, we can better test if providing more comprehensive domain-specific knowledge can help language models finish the task.
3. **The performance improvement is not significant.** Intuitively the idea should work considering more comprehensive knowledge is provided. However, according to the numbers in Table 1-3, the performance improvement is actually very limited.

In summary, the idea is elegant and promising, while the applicable scenarios and evaluation results indicate that this paper still need further revision to make it more effective.

**Questions:**

Please refer to the weakness above.

---

### Official Review · Reviewer_QtAE · 2023-11-02

**Soundness:** 3 good
**Presentation:** 3 good
**Contribution:** 3 good
**Rating:** 6
**Confidence:** 4

**Summary:**

The paper proposes using a curriculum-style prompt to improve the performance of small language models (SLMs) on mathematical reasoning tasks. ICC mirrors a curriculum learning strategy that starts with simple single-step COT examples and then increases complexity towards some multi-step examples. Their experiments on the GSM8K math reasoning dataset show ICCR outperforms the baseline COT prompt method when specializing in a 250M parameter SLM.

**Strengths:**

1. This paper introduces a sensible curriculum-based prompting strategy to address the limitations of SLMs for complex reasoning.
2. The authors compare multiple prompt designs including integration of external datasets and knowledge augmentation. Their experiments thoroughly evaluate specialization and in-context learning scenarios across model sizes.
3. The authors provide useful analysis and ablation studies to understand key factors in prompt design.

**Weaknesses:**

1. ICC prompt variations are heuristic designs lacking a formalization or learning component.
2. Evaluates on a single mathematical reasoning dataset (GSM8K) - unclear if findings generalize.
3. The knowledge augmentation (ICCKA) strategy appears ineffective but the reasons are not fully explored.
4. Qualitative analysis of differences in mistakes between small and large LMs could further inform prompt design.

**Questions:**

1. Is there a way to make the ICC prompting more adaptive or learned as opposed to pre-defined heuristic curricula?
2. Have you considered evaluating prompt designs on broader mathematical reasoning datasets besides GSM8K?
3. Could you provide examples of mistakes commonly made by SLMs vs LLMs to better understand differences in their reasoning?

---

### Official Review · Reviewer_QfrS · 2023-11-04

**Soundness:** 2 fair
**Presentation:** 2 fair
**Contribution:** 2 fair
**Rating:** 5
**Confidence:** 4

**Summary:**

This paper examines the problem of performing mathematical reasoning in small language models.
The paper proposes a type of prompt that includes a curriculum (ICCR prompt), combining prompts from tasks related to but different from the target task, ordered by their complexity.
Evaluation is based on the GSM8K dataset. While some LLMs have achieved accuracies of over 90%, small LMs such as FLAN lag substantially even with finetuning/specialization (depending on size, in the 10-30% range).
The prompting method proposed in the current paper boosts the after-specialization performance of Flan-T5-Base (up to 16.2%, compared to the baseline of 14.3%).
The paper finds in an ablation study that some aspects of the prompt design are useful.
Finally, the paper examines the usefulness of different prompts in pure ICL (no specialization); here, interestingly the curriculum design shows no benefit.

**Strengths:**

1. provides an interesting strategy crossing prompting with curriculum learning, leading to gains in the mathematical reasoning ability of a specialized small language model.

**Weaknesses:**

1. the key proposed method, described as the main contribution (at the beginning of Section 3), is only evaluated on one model (Flan-T5-Base) and dataset (GSM8K). While computational budget considerations may make it desirable to focus on the Flan-T5-Base model, evaluation on other target tasks may be a way of strengthening the generalizability of the conclusions.

2. the submission is unclear on a set of implementation points, some of them important, as described under "Questions".

3. While ICCR is framed as the key contribution, better after-specialization accuracy is in fact achieved by specializing on AsDiv IC and then using baseline prompts for inference (Table 2). On the one hand, this confirms the point that specializing on OOD prompts can be beneficial. On the other hand, it contradicts the claim that having a curriculum of different tasks, arranged in some order, is beneficial.

**Questions:**

1. Can the authors provide the full exact ChatGPT prompt (Section 3.1) in the appendix?

2. Section 3.2 "getting the highest priority and then SVAMP" -- how exactly is the different priority for each dataset specified? Also, how does this interact with the constraint that of "inclusion of at least one example from the MatHhA dataset". And how are the detailed choices motivated? Exact specification will be key, as ICCR is the key novel prompting strategy.

3. Section 3.3 "predominant operator" -- does this refer to the most frequent one?

4. Section 3.3 how exactly are the "most pertinent problems" identified? What embedding similarity metric, and what are the implementation details used for approximate nearest neirhbor search?

5. ICCKA prompts are derived based on both query and answer key. How does this work in inference, where ICCKA prompts can also be used (third row of Table 1)? Presumably the answer key should not be used in creating prompts at inference time?

6. The paper concludes from the inferior performance of ICCR-Random order that the simple-to-complex ordering of the curriculum is important. However, an alternative explanation is that ICCR-Random leads to inferior results because the ordering of the tasks in the prompt varies randomly between different prompts. What would performance be if some random ordering is chosen, and then kept constant across all prompts used in specialization?